# Vital signs-based healthcare kiosks for screening chronic and infectious diseases: a systematic review
Saksham Bhutani [1,2,3], Aymen Alian[4], Richard Ribon Fletcher[5], Hagen Bomberg[6], Urs Eichenberger[6], Carlo Menon [1,7] ✉ & Mohamed Elgendi [1,2,3,7] ✉

## Abstract

**Background** Increasing demands, such as the COVID-19 pandemic, have presented substantial challenges to global healthcare systems, resulting in staff shortages and overcrowded emergency rooms. Health kiosks have emerged as a promising solution to improve overall efficiency and healthcare accessibility. However, although kiosks are commonly used worldwide for access to information and financial services, the health kiosk industry, valued at $800 million, accounts for just 1.9% of the $42 billion global kiosk market. This review aims to bridge the research-to-practice gap by examining the development of health kiosk technology from 2013 to 2023.
**Methods** We conducted a systematic search across PubMed, IEEE Xplore, and Google Scholar databases, identifying 5,537 articles, with 36 studies meeting inclusion criteria for detailed analysis. We evaluated each study based on kiosk purpose, targeted diseases, measured vital signs, and user demographics, along with an assessment of limitations in participant selection and data reporting.
**Results** The findings reveal that blood pressure is the most frequently measured vital sign, utilized in 34% of the studies. Furthermore, cardiovascular disease detection emerges as the primary motivation in 56% of the included studies. The United States, India, and the United Kingdom are notable contributors, accounting for 43% of the reviewed articles. Our assessment reveals considerable limitations in participant selection and data reporting in many studies. Additionally, several research gaps remain, including a lack of performance testing, user experience evaluation, clinical intervention, development standardization, and inadequate sanitization protocols.
**Conclusions** This review highlights health kiosks' potential to ease the burden on healthcare system and expand accessibility. However, widespread adoption is hindered by technical, regulatory, and financial challenges. Addressing these barriers could enable health kiosks to play a greater role in early disease detection and healthcare delivery.

## Plain language summary

Our healthcare systems face increasing demands due to staff shortages, overcrowded emergency rooms, and new challenges from infectious diseases such as COVID-19. This study reviews research on health kiosks, a type of self-service technology designed to help address these challenges. Health kiosks can measure vital signs, assist in diagnosing conditions such as high blood pressure and diabetes, and offer remote care to ease the strain on healthcare workers. We reviewed studies on kiosks developed between 2013 and 2023, finding that they show potential for improving access and efficiency in healthcare. However, the widespread use of health kiosks is limited by technical, financial, and regulatory challenges. Addressing these issues could enable health kiosks to become a more common tool in healthcare, helping people access care more easily and improving health outcomes.

A global shortage of 18 million health workers is expected by 2030[1]. Currently, there are 11.89 nurses per 1000 people in the United States[2]. The ratio is even worse in China, with only 3.1 nurses per 1000 people[3]. Emergency department (ED) crowding is a major healthcare issue. Due to ED overcrowding, 10-day mortality rates were seen to increase by 50%[4]. Delays in assessing or treating patients already in the ED represent the most critical reason for ED overcrowding[5]. In its current form, the healthcare sector is slow, inefficient, understaffed, and ill-equipped. The situation is further compounded by some of the recent pandemics, such as Ebola and COVID-19, when lapses in the healthcare industry became apparent. A total of 21%

[1]Biomedical and Mobile Health Technology Research Lab, ETH Zürich, Zürich, Switzerland. [2]Department of Biomedical Engineering and Biotechnology, Khalifa University of Science and Technology, Abu Dhabi, UAE. [3]Healthcare Engineering Innovation Group (HEIG), Khalifa University of Science and Technology, Abu Dhabi, UAE. [4]Yale School of Medicine, Yale University, New Haven, CT, USA. [5]Massachusetts Institute of Technology, Cambridge, MA, USA. [6]Department for Anesthesiology, Intensive Care and Pain Medicine, Balgrist University Hospital, Zürich, Switzerland. [7]These authors contributed equally: Carlo Menon, Mohamed Elgendi. ✉e-mail: carlo.menon@hest.ethz.ch; mohamed.elgendi@ku.ac.ae

of Sierra Leone's health workforce died in the 2014-2015 Ebola outbreak[6], while 20% of all healthcare workers left their jobs since the start of the COVID-19 pandemic[7]. These facts reveal that our health force is over-burdened and exposed.

Kiosks can be an indispensable tool for addressing the growing burden on our healthcare system. Kiosks are most commonly used to provide access to information or financial services; but kiosks employed in healthcare settings are commonly referred to as "health kiosks," which generally measure vital signs, such as heart rate, blood pressure, oxygen saturation, respiratory rate, body temperature, as well as certain laboratory parameters such as hemoglobin (HbA1c) and low-density lipoprotein cholesterol (LDL). Such systems are advancing medical technology, making healthcare efficient and accessible. Health kiosks are currently employed for a variety of functions including self-check-in, telemedicine, diagnostics, and reducing ED crowding. Modern kiosks are able to help diagnose patients or provide early detection of problems ranging from hypertension and diabetes to eye and teeth diseases. The use of kiosks streamlines patient processing while reducing labor requirements and minimizes the exposure of staff to diseases while adding convenience. The ultimate goal is that utilization of health kiosks can lead to improved efficiency, accuracy, and effectiveness in early disease detection and early intervention, ultimately contributing to enhanced healthcare outcomes and patient well-being.

Global interactive kiosk market size is valued at USD 42 billion at a compound annual growth rate (CAGR) of 6% between 2022 to 2030[8]. The healthcare kiosk market is calculated at USD 0.8 billion and is expected to reach USD 1.81 billion by 2028 at a CAGR of 12.49%[9]. In 2018, sales of 432 thousand units were recorded, with the Americas as the biggest adopter, followed by Europe, Asia-Pacific, and the Middle East[10].

Two reviews, one published in 2009[11] and another in 2014[12], addressed literature predating 2013. This review focuses on literature published in the last decade. During this period, only one other systematic review on the topic has been published, conducted by Letafat-nejad et al.[13], which examines articles up until 2018. However, this review adopts a broad definition of health kiosks, encompassing various functionalities such as patient registration, feedback, information dissemination, and education. In contrast, our review is distinctly focused on a comprehensive evaluation of health kiosks designed specifically for the screening of chronic and infectious diseases through the measurement of vital signs. By narrowing our scope to this specific area of study, we aim to provide a detailed and comprehensive analysis of the effectiveness and utility of these specialized health kiosks in disease screening.

The main objective of this review is to analyze scientific publications from January 2013 to June 2023 and identify critical factors for future research. These factors include global trends, the diseases or syndromes screened by health kiosks, the vital signs monitored, the sensors used for these measurements, and the clinical outcomes of the kiosks. Additionally, the review compares the techniques employed in kiosks to validated gold standards, examines the settings in which kiosks are deployed, and evaluates methods for measuring user experience. The review acknowledges and addresses the challenges pertaining to the accessibility of health kiosks in terms of their hardware and software design, the sanitization of kiosks between users, and the regulatory considerations associated with privacy and data collection. This review also comments on the limitations of various papers and provides recommendations for future investigations.

## Methods
### Study Guidelines
This review was conducted according to the Preferred Reporting Items for Systematic Reviews and Meta-Analyses statement (PRISMA)[14]. A prior review protocol was drafted using the Preferred Reporting Items for Systematic Reviews and Meta-Analyses Protocols[15]. This review has been registered on PROSPERO [CRD42022351687]. The review protocol has been provided in Supplementary Note 1.

### Search strategy and study eligibility
The PubMed, Institute of Electrical and Electronics Engineers (IEEE Xplore), and Google Scholar were searched for articles published between Jan 1, 2013, and June 1, 2023. The detailed strategy on IEEE Xplore was the following query: *(("Full Text & Metadata":kiosk OR "Full Text & Metadata":terminal OR "Full Text & Metadata":booth OR "Full Text & Metadata":platform) AND ("All Metadata":healthcare OR "All Metadata":hospital OR "All Metadata":clinic OR "All Metadata":nursing home OR "All Metadata":primary care) AND ("All Metadata":vital sign OR "All Metadata":physiological measurement OR "All Metadata":biometric OR "All Metadata":health parameter)).* On Pubmed, the following search terms were used: *(kiosk OR terminal OR booth OR platform) AND (healthcare OR hospital OR clinic OR "nursing home" OR "primary care") AND (vital sign OR "physiological measurement" OR biometric OR "health parameter").* We included Google Scholar into our study due to its reputation as a comprehensive database, particularly for grey literature sources[16]. However, Google Scholar lacks "reliable and scalable methods to extract data"[17]. On Google Scholar the following terms were used: *(kiosk) AND (healthcare OR hospital OR clinic OR "nursing home" OR "primary care") AND ("vital sign" OR "physiological measurement" OR biometric OR "health parameter").* These search terms brought 2,160 search outcomes, but only 1000 records were displayed. Two reviews[11,12] have already addressed literature predating 2013. Moreover, this timeframe was chosen to reflect advances in smart sensors, artificial intelligence technologies, and their kiosk applications in medicine. The search for this review was completed in June 2023.

### Inclusion and exclusion criteria
In this study, we have used the following definition for healthcare kiosk: "any freestanding units containing computer programs capable of measuring vitals for the purpose of conducting disease screening." Health application software (apps) were deemed eligible for inclusion only if they were specifically designed and accessible on publicly accessible devices. Conversely, apps installed exclusively on personally owned devices, including smartphones, tablets, laptops, and desktop computers, were excluded from the study. Studies without participants were also included. Articles were excluded (a) if the focus was not on the kiosk for medical applications, (b) if it was not a health kiosk, (c) if the kiosk was not directly measuring vitals (asking users to self-report symptoms), and (d) if the article was a review article. Two reviewers (SB and ME) independently conducted the literature search, screened the titles, abstracts, and full texts for potentially eligible studies.

### Study selection and data extraction
The literature search was carried out using the Rayyan software[18]. Two authors (SB and ME) independently screened the titles and abstracts of potential studies without employing any automation tools. Any disagreements regarding the eligibility of an article were resolved through discussion. Each study deemed potentially eligible underwent full-text screening, during which study-specific information and data were extracted. Various perspectives were considered for the included articles, including the publication year, author(s), author(s)'s country, study purpose, sensor type (contact or contactless), gold standard clinical measurement, evaluation metrics, healthcare setting and adherence to design standards. Additionally, a sub-group analysis based on the purpose of the kiosks was conducted.

### Limitations assessment of individual studies
A protocol was established to assess limitations in individual studies. We selected five types of limitations to address the research questions of this review. These included: limitations in the selection of participants that could arise due to under or over-representation of participants of a particular gender, age, ethnicity, or other factors, limitations due to a small sample size of participants that could lead to chance findings, limitations due to non-response because of unwillingness or inability of participants to complete the study, limitation due to selective result reporting, and limitations in measurement of outcome due to poor performance of the system in diagnosing over dataset or compared to a clinical reference measurement. Two

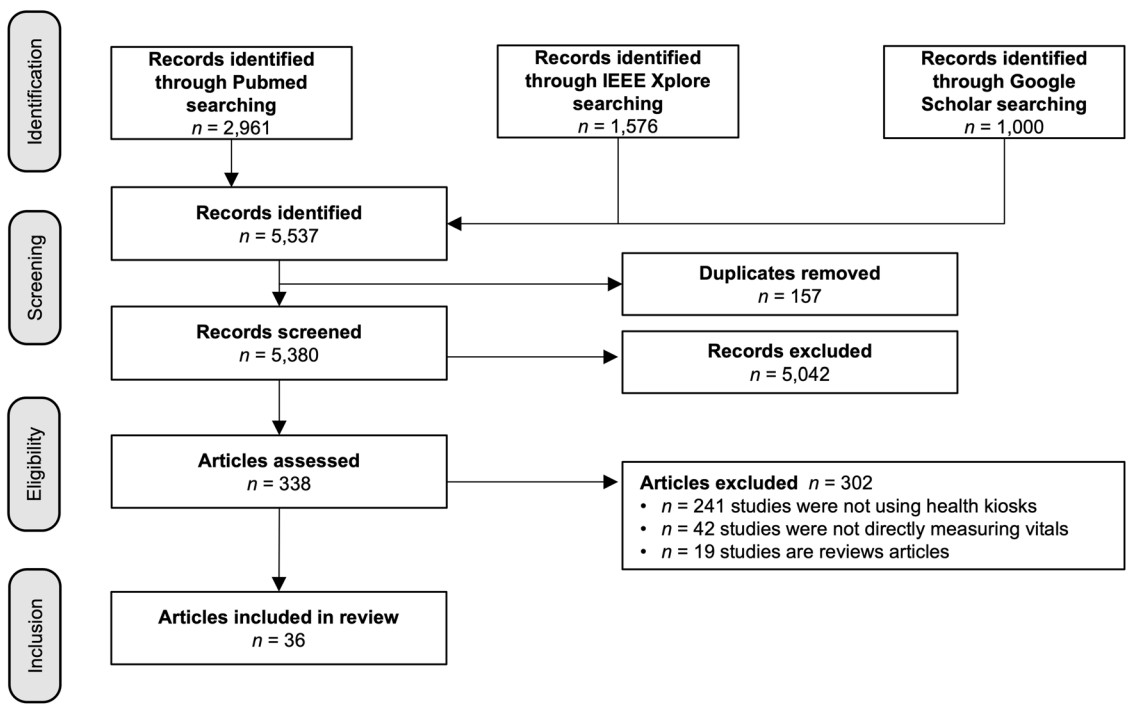

**Fig. 1 | Flow diagram of the exclusion criteria used in this study.** Out of the initial total of 5537 articles, 157 duplicates were removed. The remaining 5380 articles were screened based on their titles and abstracts, resulting in 5042 articles being deemed ineligible. Each of the 338 potentially eligible studies underwent full-text screening, leading to the exclusion of 302 studies. Ultimately, 36 studies were identified as eligible.

**Fig. 2 | Overall trends of publications using kiosks in a healthcare setting from Jan 2013 to June 2023.** The graph indicates a rise in interest in the topic, especially in 2020, which is suspected to be due to COVID-19. Data for this graph can be found in Supplementary Data 3.

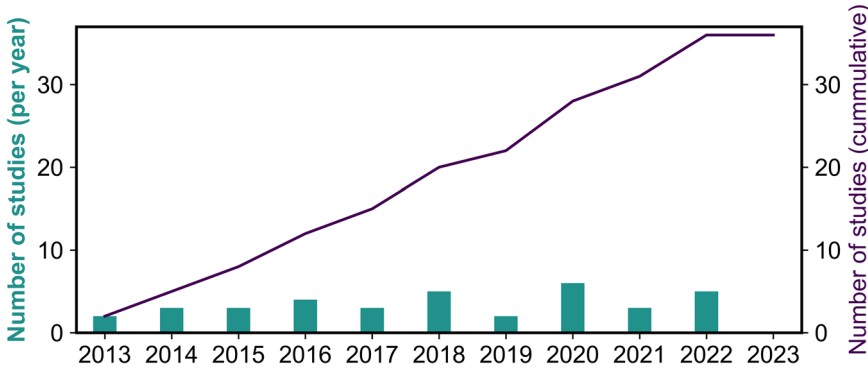

authors (SB and ME) independently assessed each study and marked it as having a high, moderate, or low level of limitation or could not be determined (due to insufficient information) for all elements. For limitations due to the small sample size, studies with less than 25 participants were deemed high limitation level, between 25 and 120 to be moderate limitation level, and more than 120 to be low limitation level. The limitation in the outcome measurement is considered a high limitation level if the error is greater than 10%, moderate limitation level if the error is between 10% and 5%, and a low limitation level if the error is less than 5%. For limitation due to non-response, if the non-response is more than 30%, then it is considered high limitation level, between 30% and 10%, then it is considered a moderate limitation level, and for less than 10%, it is considered low limitation level.

## Results
### Study selection
As shown in Fig. 1, 5537 publications were found using PubMed ($n = 2961$), IEEE ($n = 1576$), and Google Scholar ($n = 1000$) databases. We used the aforementioned search queries and applied filters to identify publications from the last ten years (January 2013 to June 2023). Studies were also excluded if they were not using health kiosks ($n = 241$), if they were review articles ($n = 19$), if they were not directly measuring vitals ($n = 42$). All the results (eligible and ineligible) are provided in Supplementary Data 2.

### Study Characteristics
All papers included in this analysis are summarized in Supplementary Data 1. Figure 2 shows the cumulative sum of publications and the total number of studies published per year that used kiosks in a healthcare setting. The findings indicate that the total number of publications on healthcare kiosks is increasing yearly. Figure 3 displays the worldwide distribution of literature pertaining to the topic under review. Notably, the United States emerged as the leading contributor with 19% of the studies, followed by India at 14%, and the United Kingdom at 10%. Most articles did not report the location of the study, in which case the locations of the institute authors are affiliated with were considered. In the case of different locations for different authors, we considered all the locations.

While health kiosks can be organized by function (e.g. triage-support, disease screening, chronic disease monitoring, etc.), it was found that the primary function of all papers reviewed was for the purpose of disease

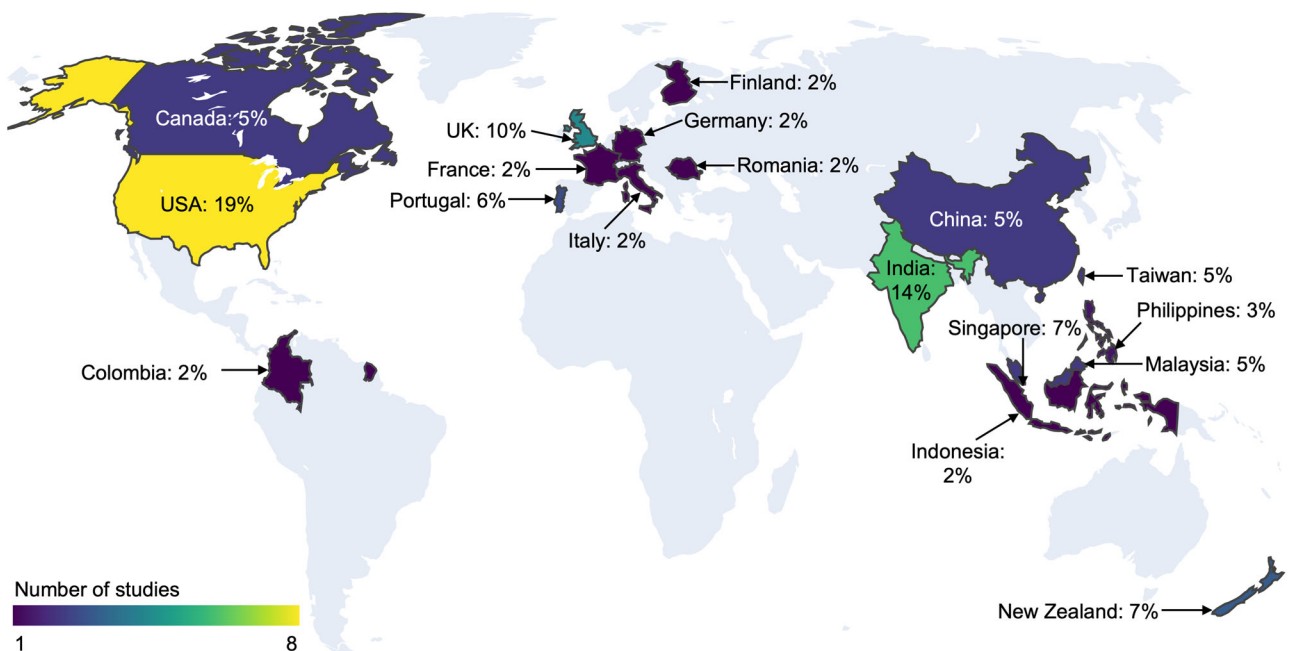

**Fig. 3 | The global distribution of scientific articles on kiosks in healthcare is based on the 36 included studies.** Detailed information about this analysis can be found in Supplementary Data 3.

screening. Therefore, we organized the papers studied in this review into six distinct disease categories, which are listed below. Some published studies failed to mention a specific purpose for the health kiosk, in which case we inferred the purpose based on the vital signs and sensors used in the study. Figure 4 shows the number of studies detecting each disease.

**Cardiovascular disease (CVD).** In total, we identified 27 studies[19–45] potentially screening for cardiovascular diseases. Gómez et al.[23] presented a kiosk for the detection and prevention of CVDs, including atrial fibrillation (AF). The kiosk recorded the blood pressure, an electrocardiogram (ECG), oxygen saturation, respiration rate, and body temperature and administered a questionnaire about the patient's family and personal history. All these parameters were fed into two machine learning models (logistic regression and random forest) to identify whether the patient had cardiovascular disease or not. After testing the kiosk on 54 participants, the authors reported the measurement error and root mean square error for all vital signs and the accuracy, F1 score, recall, precision, and specificity of the classifiers.

**Metabolic syndrome.** Seven studies[31,34,35,37,38,41,43] also monitored vitals that could be potentially used to screen for metabolic syndrome. All of these studies measured HbA1c and other hematological parameters. The HbA1c test measures the amount of blood sugar (glucose) attached to hemoglobin, and it is a good indicator of diabetes. Ng et al.[31,35] showcased a kiosk for screening chronic assessment that measured a patient's blood pressuxre, LDL-C, HbA1c, height, and weight. Next, the kiosk stratified the patients into high-, medium-, or low-limitation level categories using a simple conditional algorithm based on clinical practice guidelines. Based on the four categories, the kiosk advised the patient to refill their medicine, see a nurse, or see a doctor. Similarly, Bahadin et al.[34] proposed a kiosk for follow-up consultations for stable chronic diseases. This kiosk used blood pressure, LDL-C, and HbA1c in addition to the pulse rate. Their algorithm combined patients' physiological parameters and recent laboratory results to classify them into good, suboptimal, or poor-control groups using pre-defined rules. Based on these classifications, the kiosk produced a result slip for the patient with instructions to continue their current medications for those with good disease control or to see a nurse or doctor for further management. Finally, Liu et al.[35] designed and

optimized a healthcare kiosk to measure the patients' blood pressure, oxygen saturation, pulse rate, HbA1c, ECG, height, and weight. The authors designed an ergonomic kiosk while trying to minimize measurement errors.

**Respiratory disease.** Four articles[20,29,30,37] screened for common pulmonary or respiratory diseases. Four studies[19,20,28,29] proposed remote photoplethysmography (rPPG) kiosks that can use an RGB camera to determine the heart rate, respiratory rate, oxygen saturation, and blood pressure. These studies claimed that rPPG could be used to noninvasively measure the tissue blood volume pulses in the microvascular tissue bed underneath the skin. The kiosk by Rizal et al.[29] in particular displayed the calculated value of all the vitals and some graphs. The study validated the measurements on 11 subjects and found a mean absolute error of 1.7 Beats per minute, 0.41 Breaths per minute, and 8.15 mmHg for the pulse rate, respiratory rate, and systolic blood pressure, respectively. Pap et al.[30] presented an "eHealth Data Acquisition Kiosk" that recorded blood pressure, oxygen saturation, air flow, and galvanic skin response. The platform allowed remote data rendering for remote consultation with a physician.

**Infectious disease.** Seven of the articles[25,46–51] used the proposed kiosks to screen for infectious diseases transmitted horizontally from human to human. Notably, five of these articles[25,46–48,50] specifically emphasized the utilization of the kiosks for COVID-19 screening purposes. Khetan et al.[50] showcased the "NeelKavach" kiosk for efficiently screening people entering any premises. The kiosk used a neural network to identify whether people entering the premises were wearing masks. Each person's body temperature was then checked using a thermal camera. The data were sent to the concerned authorities if the body temperature was above a preset threshold. The kiosk also offered optional hand and luggage disinfection using an alcohol-based sanitizer and UV-C light, respectively. The authors claimed that the entire process was contactless. The kiosk was tested on 261 subjects, and an overall accuracy of 99% for the mask detection neural network was found. In another example, Ganesh et al.[25,48] introduced "AutoImpilo," an automated health machine for virtual health checkups and self-screening in rural areas. The kiosk was equipped with sensors for measuring body temperature, heart rate,

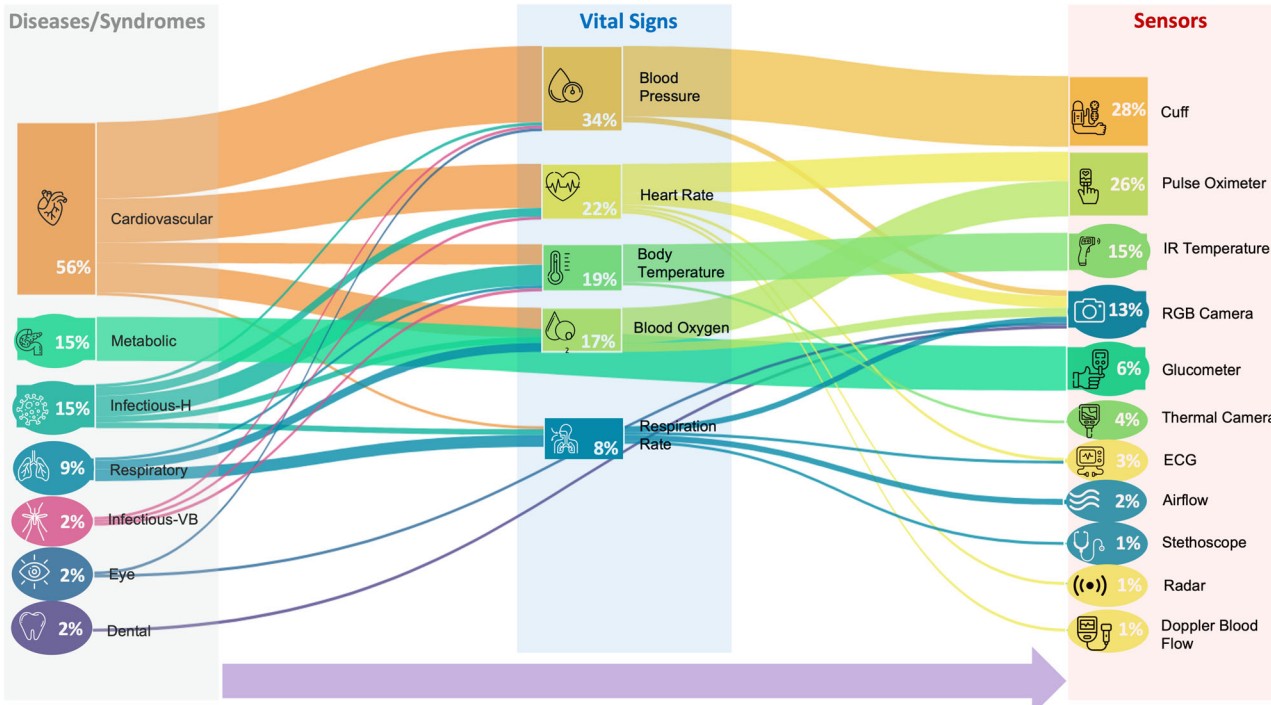

**Fig. 4 | Visualization depicting the relationship between the disease categories being-supported, the vital signs being used, and the sensors employed in the studies analyzed in this review.** Most studies (*n* = 27) detected cardiovascular diseases, followed by the detection of metabolic syndrome (*n* = 7), infectious diseases-horizontal transmission (Infectious-H) (*n* = 7), respiratory (*n* = 4), infectious diseases-vector borne transmission (Infectious-VB) (*n* = 1), dental diseases (*n* = 1), and eye diseases (*n* = 1). Blood Pressure is the most dominantly used vital sign (*n* = 29), followed by heart rate (*n* = 19), body temperature (*n* = 16), oxygen saturation (*n* = 15) and respiration rate (*n* = 7). Cuff or sphygmomanometer was the most used sensor (*n* = 24) followed by pulse oximeter (*n* = 22), infrared (IR) temperature (*n* = 13), RGB camera (*n* = 11), glucometer (*n* = 5), thermal camera (*n* = 3), electrocardiogram (ECG) (*n* = 2), airflow sensor (*n* = 2), digital stethoscope (*n* = 1), respiration radar (*n* = 1), and doppler blood flow sensor (*n* = 1).

oxygen saturation, and blood pressure. This information was then stored on a smart card that users could provide to their physician. The kiosk could also offer patients a range of common medications based on preset thresholds for the measured vitals. The authors further extended the use case of this kiosk to detect COVID-19. To this end, an infrared temperature sensor was used to determine the user's body temperature. The user's hands would then be sanitized using UV-C light if the body temperature was below the threshold. Otherwise, the kiosk would ask the user a few questions about their symptoms if the temperature was above the threshold. Based on the severity of the symptoms, the kiosk would initiate an online video call with a physician or book a COVID-19 test at a nearby hospital and an ambulance to take the patient there.

Health kiosks for infectious disease screening also included vector-borne diseases, such as malaria, which afflicts many tropical regions of the world. Magwili et al.[52] developed a kiosk specifically designed for the screening of mosquito-borne diseases in the Philippines. The kiosk effectively measures various physiological parameters such as body temperature, blood pressure, and heart rate. In addition, the system incorporates a comprehensive questionnaire that captures essential symptoms including fever, skin manifestations, vomiting, loss of appetite, and nausea, among others. To enhance the diagnostic accuracy, an expert system employing fuzzy logic and weighted rules was integrated into the kiosk, enabling the prediction of the likelihood of chikungunya, dengue, or malaria infection. Through a series of evaluations involving a total of 80 tests (20 tests per disease and 20 without), the system's preliminary diagnoses were compared against those determined by medical experts. The results revealed a remarkable level of accuracy, with the system correctly pre-diagnosing dengue, chikungunya, and malaria in 71.67%, 83.33%, and 91.67% of cases, respectively. Moreover, for diseases not classified as dengue, malaria, or

chikungunya, the system exhibited a commendable ability to correctly identify the unlikelihood of the presence of these mosquito-borne diseases, achieving an accuracy rate of 91.67%.

**Dental disease**. Pentti et al.[53] developed and tested a kiosk named the "DentoCam Extraoral Imaging System" to identify oral health care needs. The DentoCam comprises five machine vision cameras to produce five images: intraoral clinical daylight and fluorescence images of the upper and lower dental arches and daylight photographs of the front view with the posterior teeth biting together. DentoCam is accompanied by mechanical guides to help the patients assume the correct position for photographing and a live video to show the patients how visible their teeth are. The system was tested on 21 participants. A trained dentist analyzed all the images taken by the kiosk. The dentist marked all tooth fillings, missing teeth, and caries lesions. To analyze the reliability of the annotator, 13 of the images were re-annotated by a trained dentist. Cohen's kappa between the DentoCam images and the clinical evaluation was calculated to be 0.32.

**Eye disease**. One of the studies that we analyzed was that by Kapoor et al.[54], who introduced a tele-ophthalmology kiosk named "GlobeChek" for detecting common eye diseases. GlobeChek is a globe-shaped kiosk designed to perform the following screenings: intraocular pressure, pachymetry, anterior segment optical coherence tomography, posterior segment optical coherence tomography, and non-mydriatic fundus photography. The patients were categorized into one or more of the following categories based on the screenings: healthy ocular, abnormal ocular, primary glaucoma suspects, narrow-angle glaucoma suspects, diabetic retinopathy (DR), macular degeneration suspects, and other eye

conditions. Over 4 months, 326 participants were screened, of which 133 (40.79%) were detected to have a sight-threatening eye disease or condition, 47 (14.41%) had more than one disease, while the results of 192 (58.89%) came out normal. Of the 133 (40.79%) participants identified with at least one eye disease, 70 (21.47%) had primary glaucoma, 37 (11.34%) had narrow-angle glaucoma, 6 (1.84%) had DR, 4 (1.22%) had macular degeneration, and 43 (13.10%) had other eye diseases. These 133 participants were recommended to see an ophthalmologist; however, only 71 patients followed up, and the physician confirmed 33 of the 71 diseases (46.47%).

## Clinical outcomes

Ten studies[22,23,25,28,30,32,38,44,46,48] employed kiosks for telemedicine or teleconsultation, primarily in community settings with limited access to healthcare, such as rural areas. For instance, Vaidya et al.[44] tested their kiosk in rural India and found that it substantially supported the rural healthcare system through remote diagnosis. The kiosk's audio/video call feature enabled doctors to provide enhanced healthcare services to rural populations in a cost-effective manner, effectively improving healthcare accessibility and quality in these underserved regions.

Thirteen studies[20,23,25,31,33–35,37,50–54] introduced kiosks capable of assessing patient conditions or making diagnoses. These kiosks typically measured vital signs and, in many cases, also incorporated historical electronic health records or solicited additional information about the patient's current or past conditions.

Ten studies[20,23,25,33,37,50–54] developed kiosks for autonomous diagnosis. For instance, Chong et al.[37] developed a kiosk for automated triage aimed at reducing emergency department overcrowding. Their system utilized vital signs measured by sensors, along with syndrome information and the chief complaint collected through a patient questionnaire. The kiosk, employing a random forest algorithm, predicted the triage level in approximately six minutes on average, a duration comparable to manual triage times. Yao et al.[51] conducted a comparison of classification algorithms for a multimodal infection screening device based solely on vital signs. They identified support vector machines and quadratic discriminant analysis as the most effective methods, both achieving an error rate of 9.8%. Khetan et al.[50] reported faster and safer early COVID-19 detection with their kiosk, tested on over 1000 users. Gómez et al.[23] achieved F1 scores of 0.81 and 0.83 in screening cardiovascular disease risk using random forest and logistic regression classifiers, and 0.83 for arrhythmia detection using a deep neural network.

Three studies[31,34,35] focused on chronic condition assessment in healthcare settings. Bahadin et al.[34] deployed algorithms on a commercially available kiosk for chronic disease management and found the algorithms decisions in agreement with nurse clinicians (Cohen's $\kappa = 0.575$), with over 96% of patients preferring the kiosk over nurse visits. Ng et al.[31,35] reported that their kiosk's blood pressure measurements were equivalent to those by nurses, and that both physicians and patients were satisfied with the automated chronic disease care system.

Fifteen studies[19,21,24,26,27,29,36,39–43,45,47,49] utilized kiosks primarily to only measure vital signs, often integrating this data with electronic health records. Out of these five studies[21,26,29,40,41] focused on self-checkup capabilities in community setting. Rizal et al.[29] developed a kiosk that displayed calculated vital signs and graphs, validating measurements on 11 subjects with a mean absolute error of 1.7 BPM for pulse rate, 0.41 breaths per minute for respiratory rate, and 8.15 mmHg for systolic blood pressure. Three papers[21,26,40] tested commercially available kiosks in pharmacy settings to allow patients to self-check their blood pressure. These studies found the kiosks' measurements to be close to the clinical gold standard measurements.

Seven studies[19,24,27,36,39,43,47] focused on diagnostic support in a healthcare setting. Bagula et al.[36] presented a triage prioritization system that measured vital signs and employed a multivariate linear regression model to assign scores based on these signs, effectively quantifying their severity levels. This system provided a quantitative measure of their medical conditions, ensuring that the most urgent cases received timely attention. Pacheco et al.[24] developed a self-service kiosk to reduce emergency

department crowding, finding that 80% of participants found the kiosk easy to use, with younger and more educated users completing tasks faster. Brizio et al.[47] used a kiosk for automatic vitals recording during triage procedures, saving an average of nine minutes compared to conventional methods. Tompson et al.[27] evaluated a kiosk for blood pressure measurement and data upload to electronic medical records, concluding that while the system could reduce healthcare provider workload, it required higher patient utilization to be cost-effective.

## Standards of accessibility

Another critical factor that many articles overlooked is following the standards of accessibility. Making kiosks accessible requires accounting for visual, auditory, physical, speech, cognitive, language, learning, and neurological disabilities. According to the WHO[55], over 1 billion people (15% of the world's population) are expected to experience disabilities, and disabled people are far more likely to require healthcare services than people without disabilities. Thus, it is crucial to ensure that all potential kiosk users, including people with disabilities, have a decent user experience. For kiosks, accessibility standards must be followed for software and hardware design.

Some of the oldest yet relevant standards are the Americans with Disabilities Act[56] (ADA) standards for accessible design, released in 2010. The ADA, a federal civil rights law in the USA, recommends a minimum and maximum height for interactive touchpoints based on the depth of obstructions in front of the kiosk and the forward and side reach of people on wheelchairs. The ADA also requires the use of braille and the option of an audio jack with graphic applications for people with visual disabilities. Furthermore, the ADA has certain recommendations for keypads and keyboards, such as having keys with different colors and shapes to make them readily identifiable and appropriate spacings between keys. The European Union also has a law, the Accessibility Requirements for ICT Products and Services[57] (EN301-549). This law is similar to the ADA and uses the same minimum and maximum heights for interactive touchpoints. The most recent version of EN301-549, released in 2021, includes additional stipulations for font enlargement, real-time text (RTT), and visual indicators with audio, to name a few. Another comprehensive set of guidelines is the Implementation Guide Regarding Automated Self-Service Kiosks[58] by the Canadian Transport Agency. These guidelines recommend a set of minimum and maximum heights for interactive touchpoints similar to those of the ADA. Still, they also include a series of specific recommendations, such as the use of light text on the dark background for the graphical user interface (GUI), visual and audible cues for successful and unsuccessful events, and color contrast for insertion slots, among others.

The Web Content Accessibility Guidelines[59] (WCAG) focused on web content accessibility and was developed in cooperation with individuals and organizations worldwide. These guidelines apply only to software for kiosks using a web agent but are recommended to be followed by all kinds of GUIs. The WCAG has three levels for conformance: A (lowest), AA (mid-range), and AAA (highest). Level A sets a minimum level of accessibility and does not achieve broad accessibility for many situations. For this reason, it is advised to achieve the AA level. It is not recommended that Level AAA conformance be required as a general policy because it is not possible to satisfy all Level AAA success criteria for some content. AA level requires using text contrast, animations for interactions, use of simple language, minimum timeout, etc.

None of the reviewed articles mentioned whether accessibility standards were followed in their kiosks. Only three articles[31,35,38] provided the dimensions of their kiosks, two of which[31,35] were not in accordance with any accessibility standards. Thus, only one study[38] was found to follow all the accessibility standards[56–58] in terms of the hardware design. However, enough information was not provided to determine if any of the studies were following the software standards for accessibility.

## Standards for sanitization

A crucial aspect often overlooked in most studies is the sanitization of healthcare kiosks, particularly the components that users interact with

**Table 1 | Practical metrics to quantify the user experience of kiosks**

| Metric | Calculation | Purpose |
|---|---|---|
| Average Time on Task | $\dfrac{\sum_{i=0}^{n} \text{Time on Task by the } i^{th} \text{ user}}{\text{Total number of users (n)}}$ | It shows how quickly a user can complete a task. Generally, the faster users can complete a task, the better the user interface. |
| Task Completion Rate | $\left(\dfrac{\text{Number of completed task}}{\text{Total number of tasks}}\right) \times 100$ | It reflects how effectively users can complete different tasks. It can be a crucial indicator to show if tasks are clearly conveyed to the user via the interface. |
| Error Occurrence Rate | $\left(\dfrac{\text{Number of errors}}{\text{Total number of attempts}}\right) \times 100$ | It indicates how user-friendly the kiosk is. The higher the error occurrence rate, the more usability problems in the kiosk interface. |
| System Usability Scale (SUS) | Determined using a ten-item questionnaire where the contribution of each question is between 1 and 5. For odd questions, the contribution is scale position minus one, and for even questions, the contribution is five minus the scale position. Multiply the sum of a scorer with 2.5 to get SUS. | SUS, developed by John Brooke[87], is a subjective assessment of the usability of the kiosk and sub-scales of satisfaction and learnability. |
| Customer Satisfaction (CSAT) Score | Users are asked to rate their satisfaction level from 1 'very unsatisfied' to 5 'very satisfied', and then CSAT is calculated using the formula: $\left(\dfrac{\text{Number of satisfied customers (4 and 5)}}{\text{Total survey responses}}\right) \times 100$ | CSAT is a customer experience score that indicates the satisfaction level of the user. It can also indicate the value addition due to the kiosk for the user. |

directly. The surfaces of kiosks that patients interact with can harbor pathogens, facilitating the transfer of these pathogens between consecutive users. The accumulation of dirt, oils, and biological materials on sensors can further compromise the accuracy and functionality of the sensing technologies. Regulatory bodies mandate stringent sanitization protocols for medical devices to ensure patient safety and device efficacy. Consequently, it is essential for healthcare kiosks to comply with these various standards and regulations to maintain hygiene and operational integrity.

WHO's "Decontamination and Reprocessing of Medical Devices for Health-care Facilities"[60] provides comprehensive guidelines for the cleaning, disinfection, and sterilization of medical devices. The Centers for Disease Control and Prevention (CDC) in its "Guideline for Disinfection and Sterilization in Healthcare Facilities"[61] employs the spaulding classification to categorize items as critical, semicritical, or noncritical based on their risk of infection. This classification dictates the required level of disinfection or sterilization. For each of these categories the guidelines suggest disinfection or sterilization methods. The Food and Drug Administration's (FDA) "Reprocessing Medical Devices in Health Care Settings: Validation Methods and Labeling"[62], the European Union Medical Device Regulation (MDR) 2017/745[63], and the International Electrotechnical Commission (IEC) 60601-1:2024[64] mandate that manufacturers of medical devices provide comprehensive, clear, and validated instructions for the cleaning, disinfection, and/or sterilization of their devices. Specifically, the EU MDR 2017/745[63] Annex I stipulates that devices must be designed to facilitate safe cleaning, disinfection, and/or re-sterilization to prevent healthcare-associated infections.

For non-contact sensing technologies such as infrared (IR) thermometers, RGB cameras, and thermal cameras, the risk of pathogen transfer is lower, and thus no particular sanitization may be required. However, maintaining cleanliness remains crucial for ensuring accuracy. Conversely, sensors that come into direct contact with the skin necessitate slightly stricter protocols. The Association for the Advancement of Medical Instrumentation (AAMI) publishes specific standards for the cleaning and disinfection of these devices. For instance, ANSI/AAMI ST58:2018[65] provides guidelines for the selection and use of chemical disinfectants. For sphygmomanometers, the American Heart Association (AHA) recommends using disposable cuffs whenever possible[66]. The CDC advises that reusable cuffs should be disinfected with an EPA-registered disinfectant between patients to prevent cross-contamination[67].

The CDC categorizes the stethoscope as a noncritical surface and suggests that frequent disinfection with alcohol is acceptable unless the device is visibly soiled[61]. ECG leads and cables with soap and water or a disinfectant wipe between each use[61]. According to WHO, pulse oximeters must be cleaned and disinfected after each individual use and, at a minimum, weekly, prior to use on another patient[68]. They recommend wiping the device with detergent and clean water, removing any remaining detergent residue with a dry lint-free cloth, followed by cleaning with a disinfectant (as specified by the manufacturer) using a fresh cloth or disposable wipe.

Ex vivo technologies such as glucometers, which involve direct blood contact, attract more stringent safety requirements. The CDC advises against sharing glucometers between patients unless the device is designed for multi-patient use and can be properly cleaned and disinfected[69].

Notably, only one study addressed the sanitization of healthcare kiosks. Brizio et al.[47] implemented manual cleaning, wherein a staff member sanitized the kiosk between users. Additionally, six studies[19,20,28,29,49–51] utilized contactless technologies to measure vital signs. While these technologies might not necessitate frequent sanitization, maintaining cleanliness remains essential.

## Ease of use of the kiosks

Identifying objective parameters to analyze the ease of use of these kiosks is critical. Ease of use can depend on many factors, such as the placement of different interactive components of the kiosk or the clarity of instructions and language used. It becomes even more critical to quantify the user experience of the kiosks in healthcare scenarios because many require users to attach medical equipment, such as blood pressure cuffs on arms, $SpO_2$ monitors on fingertips, etc. In the book Measuring the User Experience: Collecting, Analyzing, and Presenting, Albert et al.[70] suggested different techniques for measuring user experience performance. Some techniques relevant to kiosks are time on task, the task completion rate, the error occurrence rate, the system usability scale (SUS), and customer satisfaction (CSAT). We used time on task—the time from the start to the end for using a kiosk—to compare the usability of the kiosks in the reviewed articles simply because it was the most reported value. Thirteen articles reported the total time required to use the kiosk in some form. By making multiple subjects use their kiosks, ten of the articles[20,24,28,33,34,37,47,50,53] provided the average time for the task. One of article[54] reported only the maximum time for the task, while two[32,35] mentioned both the maximum and minimum time for the task. In most studies[32,34,35,37,47,47,50,53,54], the time on task was seen to increase with an increase in the number of sensors being used. Table 1 introduces some practical metrics, methods to calculate them, and their purpose in quantifying the user experience of kiosks.

Pacheco et al.[24] developed a self-service kiosk to reduce ED crowding and thoroughly analyzed the user interface experience. The authors collected the time on task along with a questionnaire with general and system usability questions. A proportion of 80% of the participants agreed that the kiosk was easy to use. The authors also linked the time-on-task data to the

age and education level of the participants. They found that participants between 18 and 29 years of age with a university-level education took less time than the average time on task. Silva et al.[33] introduced a kiosk for patient screening and continuous monitoring. In addition to time on task and the responses to a system usability questionnaire, the authors recorded the number of interface clicks. By overlaying the location of clicks on the interface, the authors could identify whether the users were using the kiosk as per the initial design expectations.

## Discussion

The purpose of this review was to assess the current state of the literature and to guide future research dedicated to kiosks in healthcare. We sought to describe the purpose for which these kiosks are used, the setting in which they are used, the technology they use, the working principles behind them, and the future scope of these kiosks, as reported in the literature. Through our analysis, we discovered several noteworthy points related to the accessibility, user experience quantification, and testing of these kiosks, as well as future implementation challenges described below.

A common evaluation metric, Accuracy (ACC), was selected to analyze the selected studies. But, Accuracy has also been used in a different sense in different studies. Some studies report the accuracy of the prediction of a model or algorithm ($ACC_M$) which is the ratio of the number of correct predictions to the total number of input samples. Many studies report the accuracy of a sensor ($ACC_S$) which is the degree of closeness of measurements of a quantity to that quantity's true value.

We selected a definition of vital signs from each identified study to determine the key vital signs. Elliot et al.[71] claimed that acute changes in a patient's physiology can be recognized by accurately assessing their vital signs. The authors believe that the five traditional vital signs may not be adequate for detecting clinical changes in patients who have care needs that are more complex than what nurses have encountered in the past. For the aforementioned reason, the following vital signs have been considered in this paper: body temperature, heart rate, blood pressure, respiratory rate, oxygen saturation, pain, level of consciousness, and urine output. No other parameter measured in the articles was considered a vital sign. Figure 4 shows the percentage of studies using each vital sign. None of the studies used pain, level of consciousness, or urine output as vital signs. The studies evaluated in this review employed kiosks for various use cases. Multiple reviews[72–74] have explored the potential of the Internet of Things enabled platforms in enabling video consultations and tele-monitoring, thereby enabling healthcare providers to deliver essential services, during challenging pandemics such as COVID-19. This technological approach holds particular promise for aiding patients with chronic diseases in effectively managing their conditions. Vengadeshwaran et al.[22], developed a kiosk for telemedicine. In this case, the kiosk was used to facilitate a video call between the patient and the doctor, but it also measured the patient's vital signs and reported them in real-time to the doctor. This type of technology can be incremental, especially for people living in remote areas or during a crisis, such as COVID-19, when directly meeting the patient can expose the doctor to the virus. Other studies, such as that by Ganesh et al.[25], used a kiosk for end-to-end clinical visits, during which the kiosk screened the patient's vitals, diagnosed them based on these vitals, and provided them with a consultation and medicine. In yet another example, Pacheco et al.[24] used kiosks to reduce emergency room crowding. The idea was that patients could screen their vitals using kiosks and that the emergency care staff could focus on treating the patients. Kiosks can also provide a diagnostic, as shown by Rizal et al.[29] instead of going to a diagnostic center, patients can visit this kiosk and obtain a comprehensive digital report on their health.

The purposes of kiosks were also quite varied, from diagnosing patients with hypertension and diabetes to recognizing dental and eye diseases. The techniques used were also very diverse. Many of the studies relied on new-generation sensors, such as RGB cameras, and novel algorithms for measurement or diagnostics. In contrast, other studies used standardized, medically approved sensors to perform the same tasks. For example, Rizal et al.[29] used an RGB camera and developed an algorithm to measure the pulse rate, respiratory rate, and blood pressure, whereas Pap et al.[30] chose a radically different approach: using commercially available sensors to measure the same vital signs. Yao et al.[51] used a 10 GHz respiration radar to measure respiration rate, a laser Doppler blood flow meter to measure heart rate, and a thermal camera to measure the body temperature.

We selected the gold standards for measuring each vital sign based on the relevant literature and compared the performance of the techniques used in the kiosks with these values. Ogedegbe et al.[75] reported ambulatory blood pressure monitoring (ABPM) as the non-invasive gold standard for blood pressure measurements. ABPM requires the measurement of an individual's blood pressure at regular intervals for up to 24 hours. A blood pressure monitor is attached to the waist and connected to a cuff placed around the person's upper arm to be used for measurements as the individual leads their everyday life. ABPM can give a clear idea of the change in blood pressure throughout the day and also avoid problems of "white coat" syndrome, in which patients' blood pressure rises due to anxiousness about being tested. None of the 18 papers using blood pressure as a vital sign used ABPM for measurements due to the difficulty of setting up the ABPM device and the long measurement times.

For heart rate measurements, Nelson et al.[76] considered the ECG the gold standard. ECG requires attaching multiple electrodes to the patient's chest, arms, and legs to measure the electrical activity due to heart muscle depolarizations. Eighteen articles mentioned the use of heart rate as a vital sign in their kiosk. Of these eighteen articles, two articles[38,44] reported the use of ECGs for heart rate measurements, four articles[22,25,31,34] did not report the method employed, one article[29] reported the use of imaging PPG, and others used a pulse oximeter to measure this vital sign. Because ECGs are slow and require a trained medical professional, they are not ideal for kiosks.

Plüddemann et al.[77] considered invasive arterial blood gas (ABG) analysis as the gold standard for measuring blood oxygen saturation. ABG analysis requires the collection of a blood sample from an artery of the patient, followed by a measurement of oxygenated hemoglobin. Due to the invasive nature of ABG analysis, none of the kiosks studied in this review employed it. Sixteen articles[22–25,30,33,36–39,41–44,46,47,52] used pulse oximeters attached to the fingertip to measure oxygen saturation or pulse rate. Pulse oximeters use the concept of spectrophotometry, according to which oxygenated hemoglobin ($HbO_2$) absorbs more infrared light and less red light than deoxygenated hemoglobin.

Liu et al.[78] considered capnometry the gold standard for measuring the respiration rate. Capnometry requires the use of a mass spectrometer or an infrared analyzer to measure the $CO_2$ concentration in a patient's breath using an endotracheal tube (ET). Though not invasive, capnometry is a very cumbersome process that requires inserting the ET tube into the patient's trachea by a medically trained professional. For this reason, none of the seven articles using respiration rate as a vital sign used capnometry. Gómez et al.[23] utilized ECG signals to estimate the respiration rate, whereas Rizal et al.[29] used rPPG signals from the face and palm. Three studies[30,46,52] mentioned the use of an airflow sensor but did not mention the make, model, or working of this sensor. Yao et al.[51] used a Radar to estimate the respiratory rate.

Sermet-Gaudelus et al.[79] reported taking the rectal temperature using an electronic thermometer as the gold standard for measuring body temperature. However, due to sanitization issues and lubrication requirements, rectal temperatures are not used in kiosks. All the articles employed IR temperature sensors[22–25,30,37,43,46–48,52] or a thermal camera[49–51] for measuring body temperature due to the non-contact nature of this measurement.

An important parameter often overlooked in many studies was calibration. Calibration is crucial for ensuring accurate measurements. It can directly impact patient safety by minimizing the risks associated with incorrect readings. It may also aid in enhancing the longevity of the kiosk by maintaining long-term performance and reducing the need for repairs or replacements. Only four studies[20,28,43,52] addressed the calibration of sensors used in their kiosks. Of these, three studies[20,28,43] conducted calibration in controlled environments different from the actual deployment settings. Calibrating in the actual environment where the kiosk is intended to be

deployed, is essential for accounting for real-world conditions, including environmental factors such as temperature, humidity, and electromagnetic interference, as well as user influence. Only one study[52] mentioned calibration in the deployment setting, where they calibrated temperature, blood pressure, and pulse rate sensing technologies with five participants.

In terms of validation, seven studies[23,28,29,36,37,46,49] validated their kiosks in controlled conditions. Eleven studies[21,26,27,31,35,40,44,50,51,53,54] validated their kiosks in the actual settings where they were intended to be deployed. Validation is important as it ensures precision and reproducibility of the results and establishes performance benchmarks that can be used to compare different kiosks.

While conducting the literature survey, it was very difficult to perform concrete comparisons between the selected studies due to the diversity in the nature of the problems that different studies were trying to solve and the dissimilar evaluation metrics reported. It was also challenging to find a consistent definition of vital signs in order to consider all the types of measurements taken in the studies.

The studies were also categorized based on the methods used to transmit data to healthcare providers, a patient's medical home, or a prescribing provider. Sixteen studies[21–23,25,30,32,36,39,41–44,48,54] utilized the internet as the primary mode of communication, sending data from the kiosk to the cloud, where it could then be accessed by relevant parties. Telemedicine was a dominant application among these studies, with kiosks enabling patients to measure their vital signs and consult with healthcare providers remotely.

Bagule et al.[36] tested a mesh network for rural areas with unreliable internet access, where each kiosk acts as a node transmitting data via radio to other kiosks. Special relay nodes then aggregate this data and send it to the cloud using the internet. Khetan et al.[50] employed SMS messages to inform authorities about patients with a high likelihood of COVID-19 infection. Many studies[19,20,24,27,28,31,33–35,37,38,40,45,47,49,51–53] did not mention a method of communication, or their kiosks did not require communication capabilities.

The studies included in the analysis were systematically categorized based on the specific healthcare settings in which the kiosks were utilized or proposed to be used. These settings encompassed Primary Care, Secondary Care, Community, and Pharmacy. Within the Secondary Care category, further subcategories were established, namely Speciality Clinic and Emergency Department. Some studies were categorized under multiple healthcare settings, indicating their applicability in different contexts. Among the identified studies, six studies[20,30,38,42,49,51] did not explicitly specify the intended healthcare setting for their respective kiosks. Conversely, sixteen studies[19,21,25–27,31,33–36,39,41,43,45,48,52] explicitly mentioned the utilization of kiosks in a Primary Care setting. Considering the evidence presented in included studies, focused on kiosks in a Primary Care setting, it suggests a growing level of readiness for the primetime use of healthcare vital sign kiosks in clinical care. The comparative accuracy of blood pressure measurements obtained through kiosks, as assessed in multiple trials[21,35], demonstrated similar levels of error compared to clinical measurements when evaluated against the gold standard. This finding supports the reliability and feasibility of kiosk-based measurements for vital signs. Moreover, the concordance between kiosk-based diagnoses and those made by clinicians for managing patients with stable chronic diseases, as reported in other studies[34], indicates the potential of these kiosks as a valuable tool in clinical care. Furthermore, the validation of commercially available kiosk models such as PharmaSmart PS2000 and H4D ConsultStation through clinical studies[80] adds to their credibility, highlighting their readiness for use in real-world healthcare settings.

Among the studies that highlighted the use of kiosks in Secondary Care, one study specifically referred to a Speciality Clinic[53] while the remaining six studies[23,24,36,37,47,48] focused on Emergency Departments. In a thorough investigation conducted by Brizio et al.[47], involving 1844 patients in the emergency department of a private hospital in France during the COVID-19 pandemic, it was observed that the utilization of a telemedicine kiosk markedly reduced the time interval between patient registration and vital signs assessment. The study revealed that the average time saved amounted to approximately nine minutes compared to the conventional practice of nurses measuring vital signs at normal times, where the time taken was 22 minutes. This efficient and expedited process was not only found to alleviate the burden on the hospital's emergency department but also fosters a secure and trusted environment for the hospital staff and healthcare workers.

Within the Community setting, twelve studies[22,23,25,28,29,33,41,44,46,48,50,54] investigated the implementation of kiosks. Among these studies, Ahn et al.[41] conducted a test of kiosks specifically targeting the older population residing in private apartments and rest homes, providing them with the means to monitor their health conditions. Additionally, Vaidya et al.[44] developed and evaluated a mobile healthcare system designed for remote consultation with doctors, which was tested in rural areas of India. These studies contribute to the growing body of research focused on leveraging kiosk technology to enhance healthcare accessibility and provision within community settings. Finally, only three studies[25,32,40] specifically mentioned the utilization of kiosks in Pharmacy settings for telemedicine and e-diagnostics.

The use of health kiosks also raises new regulatory concerns regarding the collection and processing of personal health information, which must adhere to relevant privacy laws and regulations. For instance, in Europe, the General Data Protection Regulation[81], and in the United States, the Health Insurance Portability and Accountability Act[82], dictate the safeguarding of personal health data. Compliance with these regulations necessitates implementing measures such as secure data storage, encryption, access controls, and obtaining clear patient consent. It also requires practices like the use of pseudonyms and de-identification of personal data, ensuring individuals' rights to access their data and request its erasure, and promptly reporting any data breaches. In addition to privacy regulations, the deployment of healthcare kiosks may require compliance with specific regional or national healthcare standards. These standards ensure that the kiosks meet the necessary quality, safety, and performance criteria set forth by regulatory bodies. Moreover, the potential for incorrect or misleading information from a kiosk leading to patient harm may introduce liability concerns. Various liability laws and legal principles come into play. Medical malpractice laws may also apply if it can be demonstrated that the kiosk operator or manufacturer failed to exercise reasonable care in ensuring the accuracy and reliability of the device, resulting in harm to a patient.

The financial business model and insurance reimbursement model for the use of health kiosks remain an important challenge. This topic remains an active area of discussion for health care policy and funding, as healthcare infrastructure shifts from focusing on disease treatment to a new model that also includes disease prevention and wellness. Notably, only three studies[25,32,43] have incorporated some form of payment interface within the kiosk. Sarkar et al.[32] proposed a novel freemium model, wherein users access basic services at no cost or a minimal fee, while advanced features require payment. Additionally, they introduced an innovative payment mechanism whereby users do not pay directly at the kiosk; instead, they incur the costs when collecting their medications from the pharmacy.

The papers included in this review had certain limitations pertaining to the recruitment of the participants or data reporting. An analysis of these limitations was performed and the results are summarized in Fig. 5. An assessment of the limitations of the included studies was also performed. Figure 5a shows the limitations for the five selected elements across all the studies. Figure 5b shows the percentage of studies with low, moderate, and high limitation levels for the five selected questions. The interrater reliability suggested excellent agreement between the raters, $\kappa = 0.8$ (See Supplementary Data 4 for overall assessment while Supplementary Data 5 for detailed assessment along with the adjudication process). Limitation evaluation revealed that most studies had one or more limitations with medium or high limitation levels. Thirteen studies[19,23,25,27,29,33,35,36,38,41,45,52,53] were analyzed to have high or moderate limitation level due to the selection of participants. Twenty-seven studies[19,22–25,28–33,35–43,45,46,48,49,51–53] had high or moderate limitation levels due to the small sample size out of which eight studies[22,30–32,39,42,43,48] did not perform any testing and were assessed to have a high limitation level. Seven studies[24,27,34,45,47,53,54] had

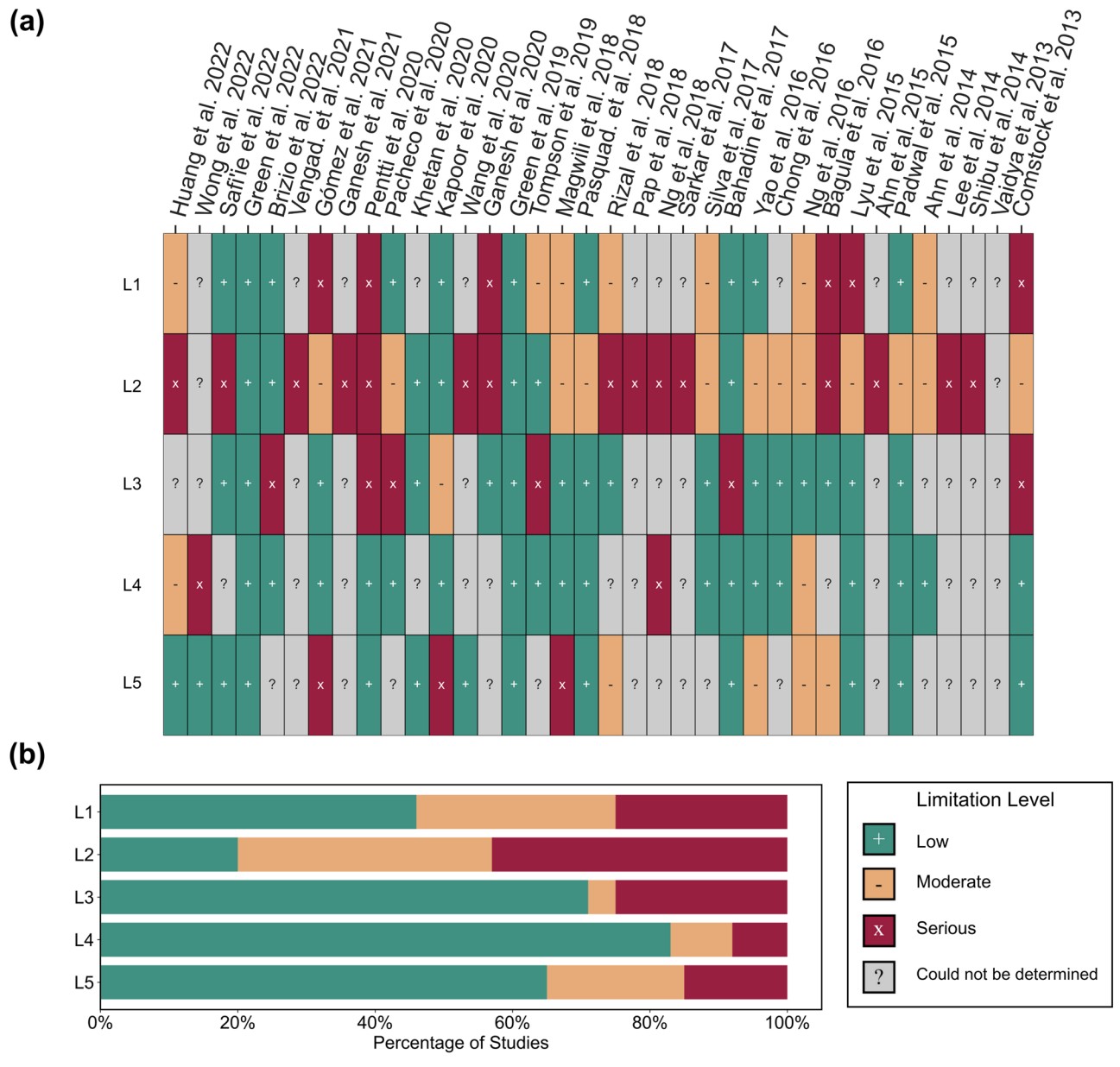

Limitations:

L1: Limitation in selection of participants (gender, age, ethnicity),

L2: Limitation due to small sample size,

L3: Limitation due to non-response (ineligibility or unwillingness of participants to complete the study or a section of the study),

L4: Limitation due to selective result reporting,

L5: Limitation in measurement of outcome.

**Fig. 5 | Subjective limitation assessment of included studies. a** Reviewers' judgments about the five elements of limitations in all included studies. **b** Reviewers' judgments about the five elements of limitations presented as percentages across all included studies. Detailed information about these analyses can be found in Supplementary Data 4 and 5.

high or moderate limitation levels due to non-response. Fifteen studies[22,23,25,29,30,32,36,39,42–44,46,48–50] did not pre-specify the analysis they are going to perform; hence, limitations in the study due to selective result reporting for these could not be determined. Eighteen studies[21,23,24,26–28,33,34,37,38,40,41,45,47,51–54] had low limitation levels, while four studies[19,20,31,35] had high or moderate limitation levels due to selective result reporting. Seven studies[23,29,35,36,51,52,54] were estimated to have high or moderate limitation levels in the measurement of outcome.

In our opinion, although the proposed kiosks show promising applications, they do not appear to be ready to become an integral part of the healthcare industry. Given the increasing interest in this topic (as shown in Fig. 2), we strongly propose the following recommendations for researchers in this field. First, all studies developing kiosks should strictly abide by accessibility standards, which need to be incorporated in the early stages of the kiosk design. These standards dictate the dimensions of the kiosk, the design and position of the interactive components, and the design of

the interface for the kiosk. We have discussed four standards from various countries across the globe in this section. Of these, we recommend following the Canadian Standards[58] for the hardware simply because these are very specific and in accordance with all other standards. For software, we recommend following the WCAG 2.1[59] AA conformance. Following accessibility standards will allow people with different disabilities, who constitute a considerable proportion of all users in healthcare settings, to use kiosks.

Next, establishing and validating comprehensive sanitization protocols is crucial for the safe operation of healthcare kiosks. Studies should develop and validate specific sanitization procedures, detailing both the methodology and frequency of cleaning. These protocols should adhere to the sanitization guidelines provided by the device manufacturers for each sensor and component within the kiosk. We have elaborated on general sanitization guidelines from various international organizations in section, which should be considered during protocol development. The kiosk's physical design should prioritize ease of sanitization, minimizing areas prone to contamination and simplifying cleaning processes. Furthermore, we encourage studies to explore automation solutions for sanitization to enhance the practicality and efficiency of real-world deployment of kiosks.

Calibration and validation of the kiosk should be performed in the deployment setting. All sensing technologies used to measure vital signs in the kiosk should be calibrated to maximize accuracy. Routine calibration is recommended to ensure consistent performance and increase the operational life of the kiosk. Additionally, the sensing technologies and algorithms used in the kiosk should be validated under real-world conditions, and the results should be published publicly. This practice will enhance reliability but also contribute to transparency and trust in the performance of the kiosks.

We recommend that the testing and reporting of the performance of kiosks should be done using standard evaluation metrics. For machine learning models used in kiosks, we recommend splitting the data between training, validation, and testing sets before training. Furthermore, the data should be fed into the model by random stratified sampling to avoid bias. We also advise that the results of the testing stage be reported as accuracy, F1 score, precision, and recall, along with the receiver operating characteristic curve for a thorough analysis of the model performance. For sensors, we recommend accuracy and precision with respect to a gold standard to be reported as evaluation metrics. Moreover, statistical analysis should be performed if the kiosk's performance is compared to that of a medical procedure. Studies can estimate metrics such as Cohen's kappa in this case.

Another important factor is to analyze the performance of the kiosks should in terms of ease of use. We have proposed five metrics to quantify the user experience of the kiosks in Table 1. Quantifying ease-of-use metrics needs to be an integral part of the kiosk design pipeline. Designers need to iterate to maximize these metrics to ensure that the kiosk is fast and easy to use for the users and efficient for the developer.

Further, integration of point-of-care testing (POCT) into kiosks' workflows is recommended to enhance diagnostic capabilities. Point-of-care tests have revolutionized diagnostic testing by enabling rapid and accurate diagnoses of specific conditions. All the kiosks discussed in this review rely on measuring physiological signals for diagnosis. Future studies should explore the utilization of POCT as an additional test when physiological signals indicate potential anomalies. Although the inclusion of POCT may introduce new challenges, such as integrating the tests with kiosk systems, adhering to more stringent sanitization protocols, and managing the disposal of testing kits, these can be addressed through meticulous planning and design.

When measuring vital signs, the kiosks should use the gold standards directly or as ground truth. Unfortunately, most of these gold standards are invasive, complex to perform, and require trained professionals. For these reasons, we recommend some practical gold standards to be used with kiosks. Sphygmomanometer (blood pressure cuff) is recommended to measure blood pressure from the patient's upper arm. The technique used is the same as mentioned above for ABPM. Measuring the blood pressure not

over 24h during daily normal life is far more pragmatic and still accurate. In a randomized controlled trial[21] comprising 510 participants, a comparison was made between blood pressure readings acquired from the commercially available PharmaSmart BP kiosk and those obtained through ABPM. The study's results indicated that the one-time readings from the kiosk exhibited a difference of 5 mmHg and 9.5 mmHg higher for systolic and diastolic blood pressure, respectively, when compared to the daytime ABPM readings. Importantly, these discrepancies fell within the acceptable error limits established by the American Heart Association[83]. Moreover, it should be ensured that the patient sits comfortably with their arm supported and that the cuff is snug to obtain an accurate reading. For the measurement of heart rate, the ECG is considered the gold standard due to its accuracy. However, the main challenge it presents is attaching the electrodes. It is difficult for untrained patients to attach these properly to the correct measured anatomical sites. One simple alternative is using fingertip-based PPG sensors, which are simpler and more practical to use and provide sufficiently similar results[84]. We recommend using pulse oximeters as practical gold standards for blood oxygen saturation measurement. This technique is not invasive and does not require medical expertise, as most other methods do. To measure respiration rate, the ECG-derived respiration (EDR) is a practical option for respiratory rate measurements. According to AL-Khalidi et al.[85], the EDR-based single-lead respiration rate is a robust method for calculating the respiratory rate. Unfortunately, due to hygiene and sanitary reasons, measuring the rectal temperature (the gold standard) is not ideal in the context of kiosks to measure body temperature. For this reason, we recommend the use of IR ear thermometers due to their markedly higher accuracy compared to other non-contact measurement techniques.

Lastly, we recommend that all studies testing kiosks on human subjects follow all ethical, legal, and regulatory norms that apply. Studies should adhere to the tenets of the Declaration of Helsinki: Ethical Principles for Medical Research Involving Human Subjects[86] and should obtain appropriate permission from the relevant bodies. We recommend that studies do not record participant data that reveal their identities. If unavoidable, studies should undertake proper measures to anonymize and dispose of the data after use. It is crucial for studies to comply with local data collection and privacy laws to ensure the lawful and responsible handling of participant information. Additionally, we strongly recommend that studies conduct comprehensive testing and validation processes to minimize the risk of liability.

This review has examined the role of kiosks in the healthcare industry based on the existing literature. Five thousand five hundred thirty-seven studies were identified from three databases, out of which 36 were selected and analyzed. The most common purpose of health kiosks was found to be cardiovascular disease screening, representing 56% of studies, with blood pressure being the most used vital sign being used in 29 of these studies. Overall, 43% of the studies were from the US, UK, and India. Most studies had considerable limitations relating to study participant selection or data reporting, with only two studies found to have minimal limitations across all criteria. The usability was good for all papers reviewed, and in general, increasing the number of sensors or vital signs was found to increase the total time required to use the kiosk. Many studies used kiosks with promising technologies like machine learning for the detection of dental disease and cardiovascular disease. Emerging technologies such as rPPG can also transform the way health kiosks are used by making the process of data collection faster, more efficient, and contactless. Based on increasing published evidence, it seems that health kiosks can help address major challenges in our healthcare infrastructure, such as the shortage of healthcare workers, overcrowding in emergency rooms, and protecting healthcare workers from unnecessary exposure to infectious diseases. It is hoped that healthcare kiosks can also greatly improve healthcare quality by increasing accessibility and efficiency by providing clinical support, reducing the need to travel, and overcoming geographic barriers. However, despite these promising applications, several key implementation challenges (technical, financial, and regulatory) currently prevent health kiosks from achieving widespread adoption as part of the standard healthcare infrastructure.

Research gaps including a lack of performance testing, user experience evaluation, clinical intervention, development standardization, and inadequate sanitization protocols remain a hurdle. Nevertheless, the long-term view of health kiosks remains optimistic, as a means of improving access to healthcare and enabling better screening and detection of common chronic and infectious diseases.

## Data availability

The protocol for this systematic review is publicly available through the International Prospective Register of Systematic Reviews (PROSPERO), with the registration number CRD42022351687. The original contributions presented in the study are included in the article/supplementary material. The comprehensive search strategy is outlined in Supplementary Note 1 (Supplementary Information). Supplementary Data 1 provides a summary of key data extracted from the selected studies, while Supplementary Data 3 presents the complete extracted data from all studies included in this review. Complete lists of all the screened publications for this study are provided as an Excel file in Supplementary Data 2. Supplementary Data 3 provides the data extracted from the studies included in this review. Data about study limitation analysis is provided in Supplementary Data 4 (overall assessment) and Supplementary Data 5 (detailed assessment along with the adjudication process). Further inquiries can be directed to the corresponding author/s.

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

## Acknowledgements
Saksham Bhutani is grateful for the support from ETH Zürich and Khalifa University (grant number RC2-2018-022).

## Author contributions
S.B. and M.E. screened and assessed articles. S.B., M.E., A.A., R.R.F., U.E., H.B., and C.M. conceived the study. C.M. and U.E. provided directions. S.B. and M.E. wrote the first draft of the manuscript. M.E. designed and led the study. A.A., H.B. and U.E. provided clinical feedback. All authors have read and agreed to the published version of the manuscript.

## Funding

## Competing interests
The authors declare no competing interests.
