## [Transparent Peer Review file · Communications Medicine]

Vital signs-based Healthcare Kiosks for Screening Chronic and Infectious Diseases: A Systematic Review

Corresponding Author: Professor Mohamed Elgendi

Version 0:

Reviewer comments:

Reviewer #1

(Remarks to the Author)

The authors conducted a systematic review of the literature surrounding use of health kiosks in care delivery. They identified papers that met their criteria for inclusion in the review, followed by thoughtful recommendations for future reviews.

Overall, I believe this topic has merit, as both prior to and all the more so after the Covid -19 pandemic more attention has been paid to the development of alternative access strategies and algorithms to improve care management.

I believe more detail is warranted in the discussion to matters related to validation of the devices and calibration in the kiosk setting, algorithms included in the models for care management, communication with the patient's medical home, or a healthcare professional or a prescribing provider, guidelines surrounding sanitization of the kiosks between users, recommendation for additional testing or point of care testing, and of course payment and clinical outcomes. These are all important elements of care redesign when considering the use of health kiosks.

I do wonder about the inclusion of the Kwok paper in which the kiosk was used to determine care management of patients with asthma in the setting of an emergency department (as opposed to screening/triage upon entering an emergency department or a kiosk in a non healthcare facility), as that analysis could be accomplished through a non kiosk based setting.

Delighted to have had the opportunity to review this important publication.

Reviewer #2

(Remarks to the Author)

Overall from your abstract the study is not inclusive based on the online database you used. Top online database such as Scopus and Web of Science was not used. These sources are mostly used in a systematic literature review study. This impacts the credibility of this study.

Reviewer #3

(Remarks to the Author)

This is an interesting review with the international focus and consideration of purpose, accessibility, usability and implementation making it relevant to a wide audience.

1. Abstract - clear, provides a good summary of the article and generally includes all pertinent information. Might be useful to add more information from conclusion, for example listing the key implementation challenges.
2. Introduction - this is a clear and succinct appropriate introduction to the article. It is confusing though that in introduction mentions only one review on topic p2 line 40 -42 (reference 11) but then under methods line 71-72 mentions recent reviews in 2009 (reference 16) and 2014 (reference 17) that have already addressed literature before 2013.
3. Review main objective lines 47-53 - Part of the objective of the review is to discuss the purposes for which kiosks are being employed. However, the article title and in the study selection and data extraction section, lines 84-90, the inclusion criteria seems to be narrower and defines healthcare kiosk as "any freestanding units containing computer programs capable of measuring vitals for the purpose of conducting disease screening." Additionally, the search includes terms for kiosk around vital signs which could have limited the literature retrieved. I think that the objectives in the introduction just needs rewording to ensure clarity and consistency throughout.
4. Appropriate that the review was conducted according to PRISMA and review protocol registered on PROSPERO. Lines

56-58. Good that PRISMA checklist is included. Would also be useful to include the review protocol as supplementary material.

5. The search is clear and generally well described. For the review appropriate databases were searched but a wider could have been search, for example CINAHL for the allied health perspective. Helpful that provides search strategies to enable replication.

6. Unsure what this means lines 108-109 - We used search queries along with search filters to identify recent publications from the last ten years (Jan 2013 to June 2023).

7. Lines 85-86 notes that 2 authors independently screen the title ad abstracts of potential studies. Please could the authors clarify how any disagreements were resolved?

8. Flow diagram – appears correct but please check wording underneath. It says out of the initial search total (n = 5,537), 301 studies were excluded, and 37 studies were identified. Need to clarify number of articles assessed or appears to have 5042 records unaccounted for.

Study characteristics are well described.

Figures 3 and figure 4 are helpful and make good use of visuals.

Accessibility is important as you highlight.

Study limitations seems appropriate, figure 5 is helpful along with the more detailed information in the supplementary material.

The conclusion summarises the article well.

Version 1:

Reviewer comments:

Reviewer #1

(Remarks to the Author)

I believe the authors have addressed the questions posed by the reviewers satisfactorily. With the many care re-design models imparted by the Covid-19 pandemic, this paper provides important data surrounding the use of kiosks as an access point.

Reviewer #3

(Remarks to the Author)

Well done on all your hard work revising the manuscript to address the extensive reviewer feedback. All of the feedback has been appropriately addressed or satisfactory explanations have been provided for not making the suggested changes. Adding the protocol as supplementary material is really useful. Additionally, the manuscript has been improved by adding the extra data and expanding the discussion to address matters related to validation of the devices, calibration in the kiosk setting, algorithms for care management, communication with healthcare professionals, guidelines for sanitization between users, recommendations for additional or point-of-care testing, payment considerations, and clinical outcomes.

Reviewer 1:

1. The authors conducted a systematic review of the literature surrounding use of health kiosks in care delivery. They identified papers that met their criteria for inclusion in the review, followed by thoughtful recommendations for future reviews.

Overall, I believe this topic has merit, as both prior to and all the more so after the Covid -19 pandemic more attention has been paid to the development of alternative access strategies and algorithms to improve care management.

Author reply: We sincerely thank you for your encouraging and detailed feedback on our manuscript. We appreciate your acknowledgment of the importance of our systematic review on the use of health kiosks in care delivery. We share your belief that the development of alternative access strategies and algorithms to improve care management has become even more crucial in the aftermath of the COVID-19 pandemic.

2. I believe more detail is warranted in the discussion to matters related to validation of the devices and calibration in the kiosk setting, algorithms included in the models for care management, communication with the patient's medical home, or a healthcare professional or a prescribing provider, guidelines surrounding sanitization of the kiosks between users, recommendation for additional testing or point of care testing, and of course payment and clinical outcomes. These are all important elements of care redesign when considering the use of health kiosks.

Author reply: Thank you for your insightful recommendations. We appreciate your detailed suggestions, which have indeed helped us enhance the manuscript. We have added data and expanded the discussion to address the matters related to validation of the devices, calibration in the kiosk setting, algorithms for care management, communication with healthcare professionals, guidelines for sanitization between users, recommendations for additional or point-of-care testing, payment considerations, and clinical outcomes.

Author action 1: Two paragraphs have been added about calibration and validation in kiosk setting to the section 4.2 “Sensing Technologies” in the discussions.

“An important parameter often overlooked in many studies was calibration. Calibration is crucial for ensuring accurate measurements. It can directly impact patient safety by minimizing the risks associated with incorrect readings. It may also aid in enhancing longevity of the kiosk by maintaining long-term performance and reducing the need for repairs or replacements. Only four studies^{20, 28, 43, 52} addressed the calibration of sensors used in their kiosks. Of these, three studies^{20, 28, 43} conducted calibration in controlled environments different from the actual deployment settings. Calibrating in the actual environment where the kiosk is intended to be deployed, is essential for accounting for real-world conditions, including environmental factors such as temperature, humidity, and electromagnetic interference, as well as user influence. Only one study⁵² mentioned calibration in the deployment setting, where they calibrated temperature, blood pressure, and pulse rate sensing technologies with five participants.

In terms of validation, seven studies^{23, 28, 29, 36, 37, 46, 49} validated their kiosks in controlled conditions. Eleven studies^{21, 26, 27, 31, 35, 40, 44, 50, 51, 53, 54} validated their kiosks in the actual settings where they were intended to be deployed. Validation is important as it ensures precision and reproducibility of the results and establishes performance benchmarks that can be used to compare different kiosks.”

Author action 2: We have added a new point in the recommendations about calibration and validation:

“Calibration and validation of the kiosk should be performed in the deployment setting. All sensing technologies used to measure vital signs in the kiosk should be calibrated to maximize accuracy. Routine calibration is

recommended to ensure consistent performance and increase the operational life of the kiosk. Additionally, the sensing technologies and algorithms used in the kiosk should be validated under real-world conditions, and the results should be published publicly. This practice will enhance reliability but also contribute to transparency and trust in the performance of the kiosks.”

Author action 3: We have updated the text under “Metabolic syndrome” heading in section 3.2 “Study Characteristics” to add more detail about the algorithms:

“... Ng *et al.*^{31,35} showcased a kiosk for screening chronic assessment that measured a patient’s blood pressure, LDL-C, HbA1c, height, and weight. Next, the kiosk stratified the patients into high-, medium-, or low-limitation level categories using a simple conditional algorithm based on clinical practice guidelines. Based on the four categories, the kiosk advised the patient to refill their medicine, see a nurse, or see a doctor. Similarly, Bahadin *et al.*³⁴ proposed a kiosk for follow-up consultations for stable chronic diseases. This kiosk used blood pressure, LDL-C, and HbA1c in addition to the pulse rate. Their algorithm combined patients’ physiological parameters and recent laboratory results to classify them into good, suboptimal, or poor-control groups using pre-defined rules. Based on these classifications, the kiosk produced a result slip for the patient with instructions to continue their current medications for those with good disease control or to see a nurse or doctor for further management. ...”

Author action 4: A new sub-section Communication Technologies (section 4.3) has been added to the discussions:

“The studies were also categorized based on the methods used to transmit data to healthcare providers, a patient’s medical home, or a prescribing provider. Sixteen studies^{21-23, 25, 26, 30, 32, 36, 39, 41-44, 46, 48, 54} utilized the internet as the primary mode of communication, sending data from the kiosk to the cloud, where it could then be accessed by relevant parties. Telemedicine was a dominant application among these studies, with kiosks enabling patients to measure their vital signs and consult with healthcare providers remotely.

Bagule *et al.*³⁶ tested a mesh network for rural areas with unreliable internet access, where each kiosk acts as a node transmitting data via radio to other kiosks. Special relay nodes then aggregate this data and send it to the cloud using the internet. Khetan *et al.*⁵⁰ employed SMS messages to inform authorities about patients with a high likelihood of COVID-19 infection. Many studies^{19, 20, 24, 27, 28, 31, 33-35, 37, 38, 40, 45, 47, 49, 51-53} did not mention a method of communication, or their kiosks did not require communication capabilities.”

Author action 5: We have added a new section “Standards of Sanitization” (section 3.5) in the results mentioning international guidelines, regulations and standards related to sanitization of the kiosk and components used inside it. We have also mentioned the sanitization in the reviewed publications.

“A critical aspect often overlooked in most studies is the sanitization of healthcare kiosks, particularly the components that users interact with directly. The surfaces of kiosks that patients interact with can harbor pathogens, facilitating the transfer of these pathogens between consecutive users. The accumulation of dirt, oils, and biological materials on sensors can further compromise the accuracy and functionality of the sensing technologies. Regulatory bodies mandate stringent sanitization protocols for medical devices to ensure patient safety and device efficacy. Consequently, it is essential for healthcare kiosks to comply with these various standards and regulations to maintain hygiene and operational integrity.

WHO’s “Decontamination and Reprocessing of Medical Devices for Health-care Facilities”⁶⁰ provides comprehensive guidelines for the cleaning, disinfection, and sterilization of medical devices. The Centers for Disease Control and Prevention (CDC) in its “Guideline for Disinfection and Sterilization in Healthcare Facilities”⁶¹ employs the spaulding classification to categorize items as critical, semicritical, or noncritical

based on their risk of infection. This classification dictates the required level of disinfection or sterilization. For each of these categories the guidelines suggest disinfection or sterilization methods. The Food and Drug Administration's (FDA) "Reprocessing Medical Devices in Health Care Settings: Validation Methods and Labeling"⁶², the European Union Medical Device Regulation (MDR) 2017/745⁶³, and the International Electrotechnical Commission (IEC) 60601-1:2024⁶⁴ mandate that manufacturers of medical devices provide comprehensive, clear, and validated instructions for the cleaning, disinfection, and/or sterilization of their devices. Specifically, the EU MDR 2017/745⁶³ Annex I stipulates that devices must be designed to facilitate safe cleaning, disinfection, and/or re-sterilization to prevent healthcare-associated infections.

For non-contact sensing technologies such as infrared (IR) thermometers, RGB cameras, and thermal cameras, the risk of pathogen transfer is lower, and thus no particular sanitization may be required. However, maintaining cleanliness remains crucial for ensuring accuracy. Conversely, sensors that come into direct contact with the skin necessitate slightly stricter protocols. The Association for the Advancement of Medical Instrumentation (AAMI) publishes specific standards for the cleaning and disinfection of these devices. For instance, ANSI/AAMI ST58:2018⁶⁵ provides guidelines for the selection and use of chemical disinfectants. For sphygmomanometers, the American Heart Association (AHA) recommends using disposable cuffs whenever possible⁶⁶. The CDC advises that reusable cuffs should be disinfected with an EPA-registered disinfectant between patients to prevent cross-contamination⁶⁷.

The CDC categorizes the stethoscope as a noncritical surface and suggests that frequent disinfection with alcohol is acceptable unless the device is visibly soiled⁶¹. ECG leads and cables with soap and water or a disinfectant wipe between each use⁶¹. According to WHO, pulse oximeters must be cleaned and disinfected after each individual use and, at a minimum, weekly, prior to use on another patient⁶⁸. They recommend wiping the device with detergent and clean water, removing any remaining detergent residue with a dry lint-free cloth, followed by cleaning with a disinfectant (as specified by the manufacturer) using a fresh cloth or disposable wipe.

Ex vivo technologies such as glucometers, which involve direct blood contact, attract more stringent safety requirements. The CDC advises against sharing glucometers between patients unless the device is designed for multi-patient use and can be properly cleaned and disinfected⁶⁹.

Notably, only one study addressed the sanitization of healthcare kiosks. Brizio *et al.*⁴⁷ implemented manual cleaning, wherein a staff member sanitized the kiosk between users. Additionally, six studies^{19, 20, 28, 29, 49-51} utilized contactless technologies to measure vital signs. While these technologies might not necessitate frequent sanitization, maintaining cleanliness remains essential."

Author action 6: A new recommendation focusing on the sanitization of the kiosks has been added.

"Establishing and validating comprehensive sanitization protocols is crucial for the safe operation of healthcare kiosks. Studies should develop and validate specific sanitization procedures, detailing both the methodology and frequency of cleaning. These protocols should adhere to the sanitization guidelines provided by the device manufacturers for each sensor and component within the kiosk. We have elaborated on general sanitization guidelines from various international organizations in section 3.5, which should be considered during protocol development. The kiosk's physical design should prioritize ease of sanitization, minimizing areas prone to contamination and simplifying cleaning processes. Furthermore, we encourage studies to explore automation solutions for sanitization to enhance the practicality and efficiency of real-world deployment of kiosks."

Author action 7: A new recommendation focusing on the additional testing of the kiosks has been added.

"Integration of point-of-care testing (POCT) into kiosks' workflows is recommended to enhance diagnostic capabilities. Point-of-care tests have revolutionized diagnostic testing by enabling rapid and accurate diagnoses of specific conditions. All the kiosks discussed in this review rely on measuring

physiological signals for diagnosis. Future studies should explore the utilization of POCT as a supplementary test when physiological signals indicate potential anomalies. Although the inclusion of POCT may introduce new challenges, such as integrating the tests with kiosk systems, adhering to more stringent sanitization protocols, and managing the disposal of testing kits, these can be addressed through meticulous planning and design”.

Author action 8: We have added more details about the payment models in the kiosk in the Kiosk Adoption and Emerging Challenges sub-section in the discussions:

“... Notably, only three studies^{25,32,43} have incorporated some form of payment interface within the kiosk. Sarkar *et al.*³² proposed a novel freemium model, wherein users access basic services at no cost or a minimal fee, while advanced features require payment. Additionally, they introduced an innovative payment mechanism whereby users do not pay directly at the kiosk; instead, they incur the costs when collecting their medications from the pharmacy.”

Author action 9: We have added a new section “Clinical Outcomes” (section 3.3) and added the following text in it:

“Ten studies^{22,23,25,28,30,32,38,44,46,48} employed kiosks for telemedicine or tele-consultation, primarily in community settings with limited access to healthcare, such as rural areas. For instance, Vaidya *et al.*⁴⁴ tested their kiosk in rural India and found that it significantly supported the rural healthcare system through remote diagnosis. The kiosk’s audio/video call feature enabled doctors to provide enhanced healthcare services to rural populations in a cost-effective manner, effectively improving healthcare accessibility and quality in these underserved regions.

Thirteen studies^{20,23,25,31,33–35,37,50–54} introduced kiosks capable of assessing patient conditions or making diagnoses. These kiosks typically measured vital signs and, in many cases, also incorporated historical electronic health records or solicited additional information about the patient’s current or past conditions.

Ten studies^{20,23,25,33,37,50–54} developed kiosks for autonomous diagnosis. For instance, Chong *et al.*³⁷ developed a kiosk for automated triage aimed at reducing emergency department overcrowding. Their system utilized vital signs measured by sensors, along with syndrome information and the chief complaint collected through a patient questionnaire. The kiosk, employing a random forest algorithm, predicted the triage level in approximately six minutes on average, a duration comparable to manual triage times. Yao *et al.*⁵¹ conducted a comparison of classification algorithms for a multimodal infection screening device based solely on vital signs. They identified support vector machines and quadratic discriminant analysis as the most effective methods, both achieving an error rate of 9.8%. Khetan *et al.*⁵⁰ reported faster and safer early COVID-19 detection with their kiosk, tested on over 1000 users. Gómez *et al.*²³ achieved F1 scores of 0.81 and 0.83 in screening cardiovascular disease risk using random forest and logistic regression classifiers, and 0.83 for arrhythmia detection using a deep neural network.

Three studies^{31,34,35} focused on chronic condition assessment in healthcare settings. Bahadin *et al.*³⁴ deployed algorithms on a commercially available kiosk for chronic disease management and found the algorithms decisions in agreement with nurse clinicians (Cohen’s $\kappa = 0.575$), with over 96% of patients preferring the kiosk over nurse visits. Ng *et al.*^{31,35} reported that their kiosk’s blood pressure measurements were equivalent to those by nurses, and that both physicians and patients were satisfied with the automated chronic disease care system.

Fifteen studies^{19,21,24,26,27,29,36,39–43,45,47,49} utilized kiosks primarily to only measure vital signs, often integrating this data with electronic health records. Out of these five studies^{21,26,29,40,41} focused on self-checkup capabilities in community setting. Rizal *et al.*²⁹ developed a kiosk that displayed calculated vital signs and graphs, validating measurements on 11 subjects with a mean absolute error of 1.7 BPM for

pulse rate, 0.41 breaths per minute for respiratory rate, and 8.15 mmHg for systolic blood pressure. Three papers^{21, 26, 40} tested commercially available kiosks in pharmacy settings to allow patients to self-check their blood pressure. These studies found the kiosks' measurements to be close to the clinical gold standard measurements.

Seven studies^{19, 24, 27, 36, 39, 43, 47} focused on diagnostic support in a healthcare setting. Bagula et al.³⁶ presented a triage prioritization system that measured vital signs and employed a multivariate linear regression model to assign scores based on these signs, effectively quantifying their severity levels. This system provided a quantitative measure of their medical conditions, ensuring that the most urgent cases received timely attention. Pacheco et al.²⁴ developed a self-service kiosk to reduce emergency department crowding, finding that 80% of participants found the kiosk easy to use, with younger and more educated users completing tasks faster. Brizio et al.⁴⁷ used a kiosk for automatic vitals recording during triage procedures, saving an average of nine minutes compared to conventional methods. Tompson et al.²⁷ evaluated a kiosk for blood pressure measurement and data upload to electronic medical records, concluding that while the system could reduce healthcare provider workload, it required higher patient utilization to be cost-effective."

3. I do wonder about the inclusion of the Kwok paper in which the kiosk was used to determine care management of patients with asthma in the setting of an emergency department (as opposed to screening/triage upon entering an emergency department or a kiosk in a non healthcare facility), as that analysis could be accomplished through a non kiosk based setting.

Author reply: Thank you for bringing this to our attention. We have re-evaluated the inclusion of the Kwok et al. paper. Upon further review, we recognize that the inclusion of this paper was ambiguous as the kiosk described in the study fits a broad definition but does not align perfectly with our specific criteria for health kiosks focused on vital signs measurement and disease screening. The analysis mentioned in the paper could indeed be conducted in a non-kiosk-based setting, and the kiosk described does not directly collect vital signs.

Based on your recommendation and a re-screening of this paper by two authors, we have decided to remove it from the review. We sincerely thank you for your advice and attention to detail, which has helped us improve the accuracy and relevance of our manuscript.

Author action: We have removed the Kwok et al. paper from the included articles. Consequently, all figures, tables, and text have been updated to reflect this change in the manuscript.

Reviewer 2:

1. Overall from your abstract the study is not inclusive based on the online database you used. Top online database such as Scopus and Web of Science was not used. These sources are mostly used in a systematic literature review study. This impacts the credibility of this study.

Author reply: Thank you for raising your concern. To optimize our search strategy, we meticulously crafted an expansive query encompassing a comprehensive set of keywords. After extensive testing across different databases, we selected IEEE Xplore and PubMed as the most relevant sources for our review. IEEE Xplore is renowned for indexing high-quality publications in engineering and technology fields, which are crucial for the technological aspects of health kiosks. PubMed focuses on biomedical and life sciences research, aligning well with our health-related objectives. Google scholar was used to capture grey literature. Google Scholar, being one of the most comprehensive databases, offers broad coverage across various disciplines and significant grey literature. This combination allowed us to capture a wide range of relevant studies.

While we acknowledge the utility of databases like Scopus and Web of Science, our selected databases provided a better alignment with our specific research requirements. Our search strategy and choice of databases were

registered on PROSPERO, ensuring a transparent and systematic approach. Our search yielded a total of 5,537 articles, which underwent rigorous screening, resulting in 37 articles that met our inclusion criteria. We conducted a comprehensive search, deliberately retaining non-peer-reviewed articles and considering studies in different languages, provided English translations were available.

Based on the editor's suggestion, we have not changed the search strategy as it has already been registered on PROSPERO.

Reviewer 3:

1. This is an interesting review with the international focus and consideration of purpose, accessibility, usability and implementation making it relevant to a wide audience.

Author reply: Thank you for your positive feedback on our review. We are pleased to hear that you find our international focus and consideration of purpose, accessibility, usability, and implementation relevant to a wide audience. Your recognition of the manuscript's potential to contribute to the field is greatly appreciated. Based on your suggestions and comments, we have subjected our work to major changes, which we feel have considerably improved the clarity and reproducibility of our work. Please find our detailed responses below.

2. Abstract - clear, provides a good summary of the article and generally includes all pertinent information. Might be useful to add more information from conclusion, for example listing the key implementation challenges.

Author reply: Thank you for your feedback. We agree that the abstract could be enhanced by incorporating more information from the conclusion.

Author action: We have updated the abstract to include key points from the conclusion:

“... Our assessment revealed significant limitations in participant selection and data reporting in many studies. Additionally, several research gaps remain, including a lack of performance testing, user experience evaluation, clinical intervention, development standardization, and inadequate sanitization protocols. Despite their promising applications, key technical, financial, and regulatory challenges currently prevent health kiosks from achieving widespread adoption as part of the standard healthcare infrastructure.”

3. Introduction - this is a clear and succinct appropriate introduction to the article. It is confusing though that in introduction mentions only one review on topic p2 line 40 -42 (reference 11) but then under methods line 71-72 mentions recent reviews in 2009 (reference 16) and 2014 (reference 17) that have already addressed literature before 2013.

Author reply: Thank you for pointing out this discrepancy. We apologize for any confusion caused by the wording. In the introduction, we intended to highlight a recent review that covers a similar timeframe (2013-2023) and explain how our review differs. In the Search Strategy and Study Eligibility section, we mentioned the two older reviews from 2009 and 2014 to justify our selection of the time frame for our review.

Author action 1: We have updated the introduction to also mention the two earlier reviews for clarity:

“Two reviews, one published in 2009¹¹ and another in 2014¹², addressed literature predating 2013. This review focuses on literature published in the last decade.”

Author action 2: We have also rephrased the sentence in the Search Strategy and Study Eligibility section for better clarity:

“Two reviews^{11, 12} have already addressed literature predating 2013.”

4. Review main objective lines 47-53 - Part of the objective of the review is to discuss the purposes for which kiosks are being employed. However, the article title and in the study selection and data extraction section, lines 84-90, the inclusion criteria seems to be narrower and defines healthcare kiosk as "any freestanding units containing computer programs capable of measuring vitals for the purpose of conducting disease screening." Additionally, the search includes terms for kiosk around vital signs which could have limited the literature retrieved. I think that the objectives in the introduction just needs rewording to ensure clarity and consistency throughout.

Author reply: Thank you for bringing this to our attention. In the main objective lines 47-53, by ‘purposes for which the kiosks are being employed,’ we meant the specific diseases the kiosk is capable of screening. We apologize for any confusion this may have caused.

Author action: We have rephrased the objectives in the introduction to ensure consistency with the inclusion criteria and the search terms used. The revised paragraph is:

“The main objective of this review is to analyze scientific publications from January 2013 to June 2023 and identify critical factors for future research. These factors include global trends, the diseases or syndromes screened by health kiosks, the vital signs monitored, the sensors used for these measurements, and the clinical outcomes of the kiosks. Additionally, the review compares the techniques employed in kiosks to validated gold standards, examines the settings in which kiosks are deployed, and evaluates methods for measuring user experience. The review acknowledges and addresses the challenges pertaining to the accessibility of health kiosks in terms of their hardware and software design, the sanitization of kiosks between users, and the regulatory considerations associated with privacy and data collection. This review also comments on the limitations of various papers and provides recommendations for future investigations.”

5. Appropriate that the review was conducted according to PRISMA and review protocol registered on PROSPERO. Lines 56-58. Good that PRISMA checklist is included. Would also be useful to include the review protocol as supplementary material.

Author reply: Thank you very much.

Author action: We have added the review protocol registered on PROSPERO as an additional supplementary file (Supplementary Information 1).

6. The search is clear and generally well described. For the review appropriate databases were searched but a wider could have been search, for example CINAHL for the allied health perspective. Helpful that provides search strategies to enable replication.

Author reply: Thank you for your insightful feedback. We appreciate your suggestion regarding the inclusion of the CINAHL database. While CINAHL offers valuable insights into nursing and allied health literature, we opted not to include it in our search strategy due to its limited scope within these specific fields and its paid access model. Our focus was on databases that provide broader and more comprehensive coverage across the diverse aspects of health kiosk technology, which led us to select IEEE Xplore, PubMed, and Google Scholar. These databases align well with our research requirements, covering engineering, biomedical sciences, and a wide range of interdisciplinary studies. Moreover, our search strategy and choice of databases were registered

on PROSPERO to ensure a transparent and systematic approach. This registration supports the replication and credibility of our review.

7. Unsure what this means lines 108-109 - We used search queries along with search filters to identify recent publications from the last ten years (Jan 2013 to June 2023).

Author reply: Thank you for bringing this to our attention. In this sentence, "search queries" refers to the search terms with Boolean operators and other data, as detailed in the Search Strategy and Study Eligibility section for each database. The "search filter" refers to the filter applied to restrict results to publications within the fixed time frame from January 2013 to June 2023.

Author action: We have reworded this sentence to improve readability:

“We used the aforementioned search queries and applied filters to identify publications from the last ten years (January 2013 to June 2023).”

8. Lines 85-86 notes that 2 authors independently screen the title ad abstracts of potential studies. Please could the authors clarify how any disagreements were resolved?

Author reply: Thank you for the question. Two authors independently screened the potential studies, and any disagreements regarding the eligibility of publications were resolved through discussion.

Author action: We have added text to clarify this in the paper:

“Any disagreements regarding the eligibility of an article were resolved through discussion.”

9. Flow diagram – appears correct but please check wording underneath. It says out of the initial search total (n = 5,537), 301 studies were excluded, and 37 studies were identified. Need to clarify number of articles assessed or appears to have 5042 records unaccounted for.

Author reply: Thank you for highlighting this. We recognize that the caption for Figure 1 was not clear. From the initial 5,537 articles, 157 duplicates were removed. The remaining 5,380 articles were screened based on their titles and abstracts, out of which 5,042 were found ineligible. Full-text screening was performed for the remaining 338 articles, out of which 301 studies were excluded, and 37 studies were finally selected. Supplementary data 1 provides the de-duplicated table of all the articles identified.

Author action: We have updated the caption of Figure 1 to clearly account for all the numbers mentioned in the figure:

“Flow diagram of the exclusion criteria used in this study. Out of the initial total of 5,537 articles, 157 duplicates were removed. The remaining 5,380 articles were screened based on their titles and abstracts, resulting in 5,042 articles being deemed ineligible. Each of the 338 potentially eligible studies underwent full-text screening, leading to the exclusion of 302 studies. Ultimately, 36 studies were identified as eligible.”

10. Study characteristics are well described.

Author reply: Thank you for acknowledging the well-described study characteristics.

11. Figures 3 and figure 4 are helpful and make good use of visuals.

Author reply: We appreciate your positive comment on the effectiveness of Figures 3 and 4.

12. Accessibility is important as you highlight.

Author reply: Thank you for recognizing the importance of accessibility highlighted in the review.

13. Study limitations seems appropriate, figure 5 is helpful along with the more detailed information in the supplementary material.

Author reply: We are grateful for your approval of the study limitations section and the accompanying figure and supplementary files.

14. The conclusion summarises the article well.

Author reply: Thank you for your positive comment on the conclusion's ability to summarize the article effectively.